# Jamestown Canyon virus is transmissible by *Aedes aegypti* and is only moderately blocked by *Wolbachia* co-infection

Meng-Jia Lau[1,2], Heverton L. C. Dutra[1,2], Matthew J. Jones[1,2], Brianna P. McNulty[1,2], Anastacia M. Diaz[1,2], Fhallon Ware-Gilmore[1,2], Elizabeth A. McGraw[1,2]*

1 Biology Department, The Pennsylvania State University, University Park, Pennsylvania, United States of America, 2 Center for Infectious Disease Dynamics, The Huck Institutes of the Life Sciences, The Pennsylvania State University, University Park, Pennsylvania, United States of America

* eam7@psu.edu

**Data Availability Statement:** All raw data can be found at https://doi.org/10.6084/m9.figshare.23677785.

## Abstract

Jamestown Canyon virus (JCV), a negative-sense arbovirus, is increasingly common in the upper Midwest of the USA. Transmitted by a range of mosquito genera, JCV's primary amplifying host is white-tailed deer. *Aedes aegypti* is responsible for transmitting various positive-sense viruses globally including dengue (DENV), Zika, chikungunya, and Yellow Fever. *Ae. aegypti*'s distribution, once confined to the tropics, is expanding, in part due to climate change. *Wolbachia*, an insect endosymbiont, limits the replication of co-infecting viruses inside insects. The release and spread of the symbiont into *Ae. aegypti* populations have been effective in reducing transmission of DENV to humans, although the mechanism of *Wolbachia*-mediated viral blocking is still poorly understood. Here we explored JCV infection potential in *Ae. aegypti*, the nature of the vector's immune response, and interactions with *Wolbachia* infection. We show that *Ae. aegypti* is highly competent for JCV, which grows to high loads and rapidly reaches the saliva after an infectious blood meal. The mosquito immune system responds with strong induction of RNAi and JAK/STAT. Neither the direct effect of viral infection nor the energetic investment in immunity appears to affect mosquito longevity. *Wolbachia* infection blocked JCV only in the early stages of infection. *Wolbachia*-induced immunity was small compared to that of JCV, suggesting innate immune priming does not likely explain blocking. We propose two models to explain why *Wolbachia's* blocking of negative-sense viruses like JCV may be less than that of positive-sense viruses, relating to the slowdown of host protein synthesis and the triggering of interferon-like factors like Vago. In conclusion, we highlight the risk for increased human disease with the predicted future overlap of *Ae. aegypti* and JCV ranges. We suggest that with moderate *Wolbachia*-mediated blocking and distinct biology, negative-sense viruses represent a fruitful comparator model to other viruses for understanding blocking mechanisms in mosquitoes.

**Funding:** This research was supported by the R01 AI 143758 from the National Institutes of Health (nih.gov) to EAM. The funders played no role in the study design, data collection and analysis, decision to publish or preparation of the manuscript. EAM received summary salary support from the grant.

**Competing interests:** The authors have declared that no competing interests exist.

## Author summary

Jamestown Canyon virus (JCV), a newly emerging virus in North America, can result in disease spillover from wild mammals into human populations via the bite of infected mosquitoes. We show that the mosquito *Aedes aegypti*, known for transmitting many viral pathogens to humans globally, and whose distribution is creeping northward in the USA toward regions where JCV is present, is likely able to transmit the virus. *Wolbachia* is an endosymbiotic bacterium being released in wild mosquito populations of because it limits the replication of human viruses inside the mosquito, limiting their transmission to humans. We show that *Wolbachia* has a limited ability to control the replication of JCV, which is likely because *Wolbachia*-induced antiviral response is quite weak, and unique aspects of negative-sense virus biology make them less susceptible to blocking. Our findings suggest that JCV may serve as a comparative model to positive-sense viruses like dengue in dissecting the mechanism of *Wolbachia*-mediated virus blocking. It also warns that shifting mosquito distributions, as expected under a changing climate, could bring JCV and Aedes mosquitoes into greater contact, potentially increasing the incidence of JCV in humans.

## Introduction

Jamestown Canyon virus (JCV) is a negative-sense arbovirus in the genus *Orthobunyavirus* (family *Peribunyaviridae*), and a relative of La Crosse (LACV) and California Encephalitis viruses. JCV is part of an enzootic cycle involving White-tailed deer, as well as other ungulates, (reviewed in [1]). Transmitted between mammals by mosquitoes, JCV has been found in numerous species including at least 22 members [2] of the genera *Aedes*, *Culex*, *Culiseta*, *Ochlerotatus*, and *Coquillettidia* (reviewed in [3]). Currently, the upper Midwest of the USA, including the states of Minnesota and Wisconsin, has the highest incidence of human JCV [4]. Seroprevalence in humans in some communities has been found to be as high as 20–40% [2]. Infection is usually asymptomatic but can induce fever, malaise, and headache [1]. Unlike most arboviral illnesses, JCV can also cause respiratory symptoms. In rare instances, the disease becomes neuroinvasive, with death occurring in $< 2\%$ of cases [5]. JCV in humans has been on the rise over the last few decades, exhibiting a $> 9$-fold increase in symptomatic cases. The increase may be explained by better detection and awareness [3,6] as well as factors known to drive enzootic disease spillovers like the destruction of habitat, human encroachment, and climate change [2].

*Ae. aegypti*, and to some extent its sister species *Aedes albopictus*, transmit some of the most prevalent arboviruses globally including dengue (DENV), Zika, chikungunya, and Yellow Fever [7]. The distribution of both vector species has expanded in recent decades, from tropical to subtropical and even to some temperate regions around the world [8,9]. By 2050, the more tropical *Ae. aegypti* is expected to occur consistently in the upper Midwest [8], where there are already sporadic reports during the summer months [4]. In addition, both case and seroprevalence data for JCV in humans and animals suggest the virus is present beyond the upper Midwest, including in the Western, Northeastern, and Southern regions of the USA [3]. To date, JCV has not been isolated from field-caught *Ae. aegypti* or *Ae. albopictus* and there have been no laboratory studies to test for vector competence.

Without deployable vaccines for many of the above viruses, vector control has been the primary means of limiting arboviral diseases in human populations. A recent and promising approach has involved the use of the insect bacterial endosymbiont, *Wolbachia* as a biocontrol

agent [10]. *Wolbachia* infection in insects has the effect of limiting the ability of any co-infecting positive-sense viruses to replicate. This effect combined, with the vertically transmitted microbe's ability to spread through populations has rendered it a powerful agent against virus transmission. *Wolbachia's* release into *Ae. aegypti* populations in the field have conferred substantial reductions in dengue fever incidence in human populations living in the release zones in Malaysia, Brazil, and Indonesia [11–13]. Since the discovery of *Wolbachia*-viral mediated blocking, its mechanistic basis has been highly sought after but remains incomplete. Understanding the basis of the blocking trait is a priority for the effective rollout of this agent across a diverse global landscape and into the future where virus [14] or mosquito [15] evolution may challenge the effectiveness of blocking. The current consensus is that the trait is likely multifaceted (reviewed in [16]), involving diverse aspects of insect physiology, part of which includes the insect immune response, triggered by the bacterial endosymbiont. Specific antiviral pathways and more generalist antioxidant responses [17–19] are part of this 'innate immune priming' response. *Wolbachia's* ability to block positive-sense RNA viruses including DENV, Zika, chikungunya, and Yellow Fever has been well characterized [20–22]. In contrast, blocking of the negative-sense viruses LACV, vesicular stomatitis virus (VSV), and cell-fusing agent virus has only been tested *in vitro*, with little evidence of blocking [23,24]. With a poor understanding of whether blocking is mediated locally at the level of the cell, systemically [25,26], or both, vector competence experiments in the whole mosquito are needed for negative-sense viruses. Comparing the virus-blocking ability of *Wolbachia* with different types of viruses may offer a means to further dissect the blocking mechanism.

The innate immune landscape of a mosquito is shaped independently by *Wolbachia* and viral infections, and by their interactions during co-infection. The RNAi and the Toll pathways are considered the most important immune responses against positive-sense DENV in mosquitoes [27,28]. In *Drosophila* RNAi has also been shown to be critical for limiting negative-sense viruses [29]. In mosquitoes, the antiviral JAK-STAT pathway, activated by Vago protein [30][30], an interferon-like factor, is necessary for controlling DENV [31]. The activation of Vago through another pathway, IMD, is likely a means by which *Wolbachia* reduces loads of co-infecting positive-sense viruses [32,33]. *Wolbachia* infection triggers the expression of both Toll and IMD pathways by inducing reactive oxygen species (ROS) in mosquitoes [17]. Interestingly, in mammals, interferons represent one of the key host defenses against infection with negative RNA viruses, like Rift Valley fever virus [34].

We carried out a vector competence study of JCV in wildtype and *Wolbachia*-infected *Ae. aegypti* to understand the potential emergence of this mosquito species as a vector, and to further explore the limits of *Wolbachia*-mediated blocking. We also examined the innate immune gene expression of *Ae. aegypti* in response to JCV, *Wolbachia*, and co-infection with both, to assess the reaction of the non-native vector to this new virus and the potential role of immunity in blocking negative-sense viruses.

## Material and methods

### Mosquito rearing

We used two mosquito lines in our experiments. A wildtype, *Wolbachia*-free line derived from a lab-established colony of wildtype mosquitoes collected ~5 years prior in Mérida, Mexico. The second line consisted of a laboratory colony of *Ae. aegypti* infected with the *w*AlbB strain of *Wolbachia* [35] that was backcrossed into the Mérida line. All mosquitoes were maintained in a climate-controlled insectary, at 26 ± 1˚C and 68 ± 5% relative humidity, with a 12:12 hour light: dark cycle. Larvae were reared in plastic trays (40 × 30 × 8cm) containing deionized (DI) water at a density of 300 larvae/3L of water and fed with one tablet of Tropical Fish Food

(Tetramin: Tetra Werke, W. Germany) every two days. Adult mosquitoes received 10% sucrose solution *ad libitum*, and females were blood-fed weekly with human blood from anonymous donors (BioIVT: Westbury, NY) using a Hemotek membrane feeder (Hemotek Ltd., UK) when eggs were required for experimental or colony rearing purposes.

### Infection of mosquitoes with Jamestown Canyon virus

Vero cells were cultured at 37˚C (5% $CO_2$) with Dulbecco's Modified Eagle Medium (DMEM) containing 10% FBS (Fetal bovine serum) and 1% Penicillin-Streptomycin (Pen-Strep). The Jamestown Canyon virus (strain CT 15-8-72591 –passage history = Vero 2) was obtained from the UTMB Arbovirus Reference collection. When cells reached 80% confluency, the medium was removed and replaced with new media containing 2% FBS. Cell culture supernatants were collected 3 days post-infection, and the number of virus gene copy numbers was determined through RT-qPCR. Virus-infected cell supernatant was then re-suspended 1:1 in fresh whole human blood and used in an infectious blood meal. 6 ± 1 day-old adult mosquitoes were immobilized on ice and females were spread into 2 L paper containers (Zoro, Cat. no. G4215843) and were starved for ~18 hours before feeding. Females were fed on the blood-virus mixtures or blood-mock mixtures (1: 1) for 40 minutes using the Hemotek membrane feeder system to keep the blood warm at 37˚C. After feeding, females were immobilized on ice the second time, and engorged females were sorted into 16 oz paper containers (Zoro, Cat. no. G4141713) and provided with 10% sucrose solution ad libitum through a mesh lid.

At 0 dpi, whole females from both the *Wolbachia* infected or uninfected lines were collected in 2 mL micro tubes containing a ceramic bead and 300 μL of TRI Reagent (Sigma-Aldrich, Cat. no. T9424). These individuals were then used to assess initial viral load post-feeding (S1 Table), with individuals averaging ~$10^6$ viral copies per individual. The remaining fully engorged females were separated into containers for collection at 3, 7, 10, and 14 days post-infection (dpi). In the first replicate, at each time point we collected samples of females from four tissues: wings + legs, saliva, abdomen and head + thorax. The saliva samples were first collected using a previously established capillary method [21]. All samples were stored in 300μL of TRI Reagent, kept at -80˚C until further processing. In the second independent replicate, we only tested abdomen and head + thorax. Infection rates (virus prevalence) in the abdomen were explored to capture early infection events, including virus in the midgut tissue, whereas head/thorax measures served as proxies for dissemination. Legs and saliva measurements are used to indicate the potential transmissibility of virus and the result from legs might be more accurate [36]. Viral loads (viral copy numbers) were measured to quantify strength of infection.

### Jamestown Canyon virus quantification

For nucleic acid extraction, samples were homogenized as previously described [37] using a Bead Ruptor Elite (Omni International, Kennesaw, GA). Trizol (Invitrogen) extraction of total RNA from individual whole mosquitoes was performed following manufacturer's instructions, resuspended in nuclease-free water, and quantified using the NanoDrop 2000 spectrophotometer system (ThermoFisher Scientific). Each RNA sample was then diluted to 20 ng/μL in 20 μL nuclease-free water and stored at -80˚C. JCV levels per tissue were quantified by RT-qPCR using a LightCycler 480 instrument (Roche). The primer set was designed by Kinsella et al [5] and quantified by SYBR Green assay (S1 Text).

### *Wolbachia* qualification and gene expression assays

After infectious blood feeding, whole-body *Wolbachia* positive and negative females were collected at 3, 7, 10 and 14 dpi, stored in 300μL of TRI Reagent, kept at -80˚C for confirmation of

their total body virus loads DNA and RNA were extracted from all samples collected above using a Direct-zol DNA/RNA Miniprep kit (Zymo Research, Cat No. R2080), and RNA was used for gene expression, while DNA was used for *Wolbachia* quantification followed the methods described previously [1]. Relative *Wolbachia* density was based on a $2^{-\Delta Ct}$ method using mosquito (RPS17 forward: 5'-TCCGTGGTATCTCCATCAAGCT-3', reverse: 5'-CAC TTCCGGCACGTAGTTGTC-3') [15] and *Wolbachia* Walb (gene *WD0513*) [38] forward: 5'-CCTTACCTCCTGCACAACAA-3', reverse: 5'-GGATTGTCCAGTGGCCTTA -3') primer sets that described previously. The qRT-PCR cycle was activated at 95˚C for 30 seconds (Ramp Rate: 4.4˚C/s), followed by 40 amplification cycles and melting curve analysis same as the viral quantification assays described (S1 Text).

To test the immune response of female *Wolbachia* positive and negative *Ae. aegypti* after Jamestown Canyon virus infection, we selected four genes: *MYD88*, *IMD*, *hopscotch* and *AGO2* to represent the immune response from the Toll pathway [17], IMD pathway [17], JAK/STAT pathway [31,39], and RNAi pathway [40] respectively (Table 1). In addition, we also tested the expression level of *AeVago1*, which encodes Vago protein, an interferon-like factor. The qPCR was run under the same conditions as above for virus quantification. We used gene *rps17* to normalize expression levels through the $2^{-\Delta\Delta Ct}$ method [41]. Primers used are listed in S2 Table.

## Female longevity

The longevity of *Aedes aegypti* females was tested under the infection of *Wolbachia* and/or JCV virus (that fed with JCV or its mock media). After feeding, engorged females were immobilized on the ice and isolated in groups of 14–15 females in 16 oz containers. Each line was represented by eight to eleven cups. Mosquitoes were fed through mesh lids using 2 cotton balls, one soaked in 10% sucrose and another with only water, that were changed every 2 days. Female survival was checked every two days from 4 to 66 days post-infectious feeding.

## Statistical analysis

We used R v. 4.2.1 to conduct data analyses and visualizations. For virus quantification and dissemination, we used a generalized linear model to test the infection prevalence in different tissues and a generalized additive model to test the viral load after JCV infectious blood feeding. The relative gene expression and *Wolbachia* density data were log 10 transformed for ANOVA analysis. Longevity data was compared using the Log-Rank survival test. We also carried out pair-wise comparisons between *Wolbachia* infected and uninfected strains using Mann-Whitney-U-Test (for viral load), Chi-squared test (for infection prevalence) or Tukey's HSD test (gene expression).

# Results

## Vector competence for Jamestown Canyon virus

We infected mosquito lines with and without *Wolbachia* infection with Jamestown Canyon virus (JCV) by oral feeding and then studied viral prevalence and viral load across key tissues

**Table 1. Current evidence from the literature on role of immune genes in response to *Wolbachia* and RNA virus infection.**

| Gene ID | pathway | Responds to *Wolbachia*? | Inhibits positive sense RNA viruses? |
|---|---|---|---|
| *MYD88* (AAEL007768) | Toll pathway | Yes [18] | Yes [28] |
| *IMD* (AAEL010083) | IMD pathway | Yes [18] | unknown |
| *hopscotch* (AAEL012553) | JAK/STAT pathway | unknown | Yes [31] |
| *AGO2* (AAEL017251) | RNAi pathway | unknown | Yes [42] |
| *AeVago1* (AAEL000200) | encode Vago protein | Yes [32] | Yes [32] |
| *rps17* (AAX84650) | 40S ribosomal protein S17 | Reference gene | |

(abdomen, head & thorax) at several days post-infection (dpi). The *Wolbachia*-free (wildtype line) showed very strong viral prevalence in mosquito tissues in both replicate experiments, with rates >90% by three dpi (Figs 1A and S1A), in both the abdomen and head & thorax, indicating dissemination of virus. Viral loads averaged across all dpi and both replicate experiments were high; $5.25 \times 10^{7}$ copies/tissue in abdomen and $8.39 \times 10^{7}$ copies/tissue in the head & thorax (Fig 1B). Viral loads were also highly bimodal, a common feature of viral loads in mosquitoes [43]. Virus prevalence in legs and saliva (Fig 2A) were near 100% across all dpi, and above 50% after dpi 7, respectively. Saliva estimates of viral prevalence are a conservative estimate of transmissibility [36]. Average viral loads in the legs and saliva (Fig 2B) were $6.84 \times 10^{7}$ and $1.00 \times 10^{4}$ copies/tissue, respectively. These data strongly demonstrate that, *Ae.*

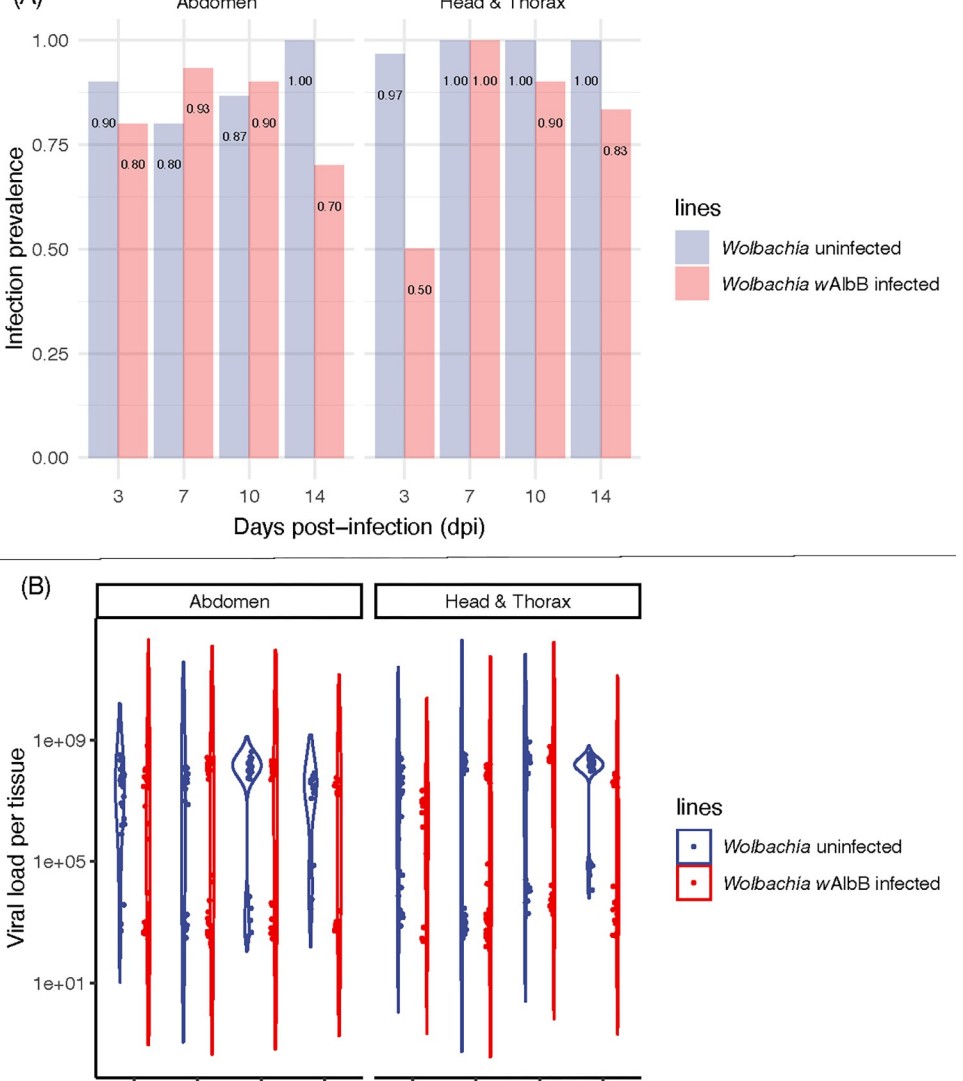

**Fig 1.** (A) Jamestown Canyon virus (JCV) infection prevalence (proportion infected) in adult female mosquito abdomen and head & thorax at 3, 7, 10 and 14 days post-infection (dpi) for replicate experiment 1. (B) Quantification of JCV load (viral copy number) in adult female mosquito abdomen and head & thorax at 3, 7, 10 and 14 days post-infection (dpi) for replicate experiment 1. $n$ = 24–30 per treatment.

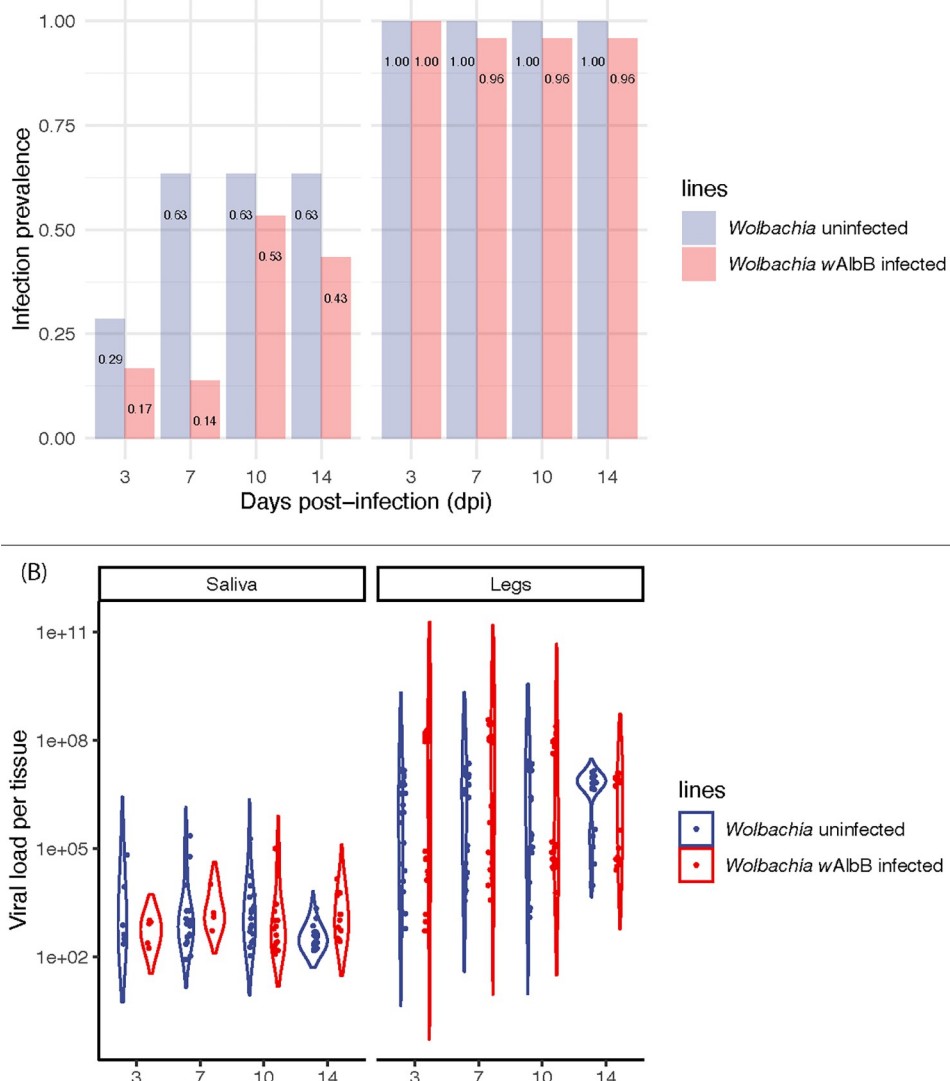

**Fig 2.** (A) Jamestown Canyon virus (JCV) infection prevalence (proportion infected) in adult female mosquito saliva and legs at 3, 7, 10, and 14 days post-infection (dpi) for replicate experiment 1. (B) Quantification of JCV load (viral copy number) in adult female mosquito abdomen and head & thorax at 3, 7, 10, and 14 days post-infection (dpi) for replicate experiment 1. $n$ = 17–30 per treatment.

*aegypti* is highly competent for JCV, allowing rapid virus dissemination through the mosquito body and excretion into the saliva.

The effect of *Wolbachia* infection on JCV prevalence (Figs 1A and S1A) varied by tissue but was consistent across two replicate experiments (replicates: z = -1.29, p = 0.20; *Wolbachia* infection: z = -5.65, p < 0.001; dpi: z = 5.68, p < 0.001, tissue: z = 1.79, p = 0.074). There was no effect of *Wolbachia* infection in abdomen (z = -1.14, p = 0.25), nor an effect of dpi (z = -0.09, p = 0.93). There was, however, a significant interaction (z = 4.11, p < 0.001), whereby *Wolbachia* reduced prevalence at dpi 3 and 14 in replicate one (Fig 1A) and dpi 3 (S1A Fig) in replicate two. For head & thorax, *Wolbachia* infection (z = -3.32, p < 0.001), dpi (z = 2.99, p = 0.003), and the interaction (z = 3.19, p = 0.001) were all significant. Like in abdomen,

*Wolbachia* infection showed the greatest ability to reduce the prevalence of JCV at 3 and 14 dpi (Fig 1A) in replicate one, and 3 dpi in replicate two (S1A Fig). When *Wolbachia* infection was associated with reduced prevalence, reductions ranged from 20–50%.

The effect of *Wolbachia* on viral load (Figs 1B and S1B) varied across the two replicate experiments (replicate: t = 4.49, p < 0.001; *Wolbachia* infection: t = -0.48, p = 0.63; dpi: t = 1.74, p = 0.082, tissue: t = 3.83, p < 0.001). We, therefore, analyzed the replicates separately. For replicate one (Fig 1B), *Wolbachia* had no effect in the abdomen (t = -0.68, p = 0.49), nor dpi (t = - 0.67, p = 0.51), but had a significant interaction (t = 4.80, p < 0.001). Specifically, *Wolbachia* led to decreased loads in JCV at dpi 14 in replicate 1 and day 3 in replicate 2 (S3 Table). For head & thorax, *Wolbachia* infection (t = -2.34, p = 0.020), dpi (t = 3.04, p = 0.002) and the interaction (t = 2.57, p = 0.011) were all significant. *Wolbachia* infection reduced viral loads on average by a factor of 0.68 across all time points in the head & thorax, with the greatest effect at day 3. In contrast, for replicate two (S1B Fig), in either tissue *Wolbachia* (abdomen: t = 0.22, p = 0.82; head & thorax: t = 1.94, p = 0.053) or dpi (abdomen: t = 1.84, p = 0.066; head & thorax: t = - 0.97, p = 0.33) had no effect on viral load, but a significant interaction was found for head & thorax but not for abdomen (abdomen: t = 0.22, p = 0.83; head & thorax: t = 2.67, p = 0.008). Like replicate 1, the greatest reduction in JCV in association with *Wolbachia* infection was at dpi 3 (S3 Table). In summary, the ability for *Wolbachia* to control JCV is most pronounced both early and late during infection. This may suggest an ability to both delay early replication as well as speed clearance.

Saliva and legs were tested in addition to measure JCV dissemination efficiency, an approximation of viral transmissibility. The prevalence of virus in saliva was affected by *Wolbachia* infection (z = -3.66, p < 0.001) and dpi (z = 3.57, p < 0.001), and with a significant interaction (z = - 2.36, p = 0.018). *Wolbachia* infection was more likely to reduce viral load in the early time points (Fig 2A). In legs there was no effect of *Wolbachia* infection (z = - 0.01, p = 0.995) or dpi (z = - 0.78, p = 0.43). For viral load in saliva (Fig 2B), there was no effect of dpi (t = - 1.03, p = 0.31) or *Wolbachia* infection (t = - 0.69, p = 0.49). Viral load in legs (Fig 2B) was significant for both dpi (t = - 3.64, p < 0.05) and *Wolbachia* infection (t = 6.07, p < 0.05). On average, across all time points, *Wolbachia* infection only reduced infection prevalence by 0.23% in saliva. Viral load in saliva was reduced by ~50% on average (*Wolbachia*: $5.06 \times 10^3$ copies/tissue; uninfected: $9.62 \times 10^3$ copies/tissue) but unexpectedly increased it by ~9.5 times on average in legs (*Wolbachia*: $3.28 \times 10^7$ copies/tissue; uninfected: $4.72 \times 10^6$ copies/tissue).

## Effect of Jamestown Canyon virus infection on immune gene expression

We selected five mosquito immune genes whose expression has been shown to control virus infections and/or be responsive to *Wolbachia* infection. We then tested their relative expression levels in *Wolbachia w*AlbB infected and uninfected lines, with and without JCV infection at different dpi (Fig 3). Mosquito samples from the same line fed with mock-infected blood served as controls. Although the response of these pathways varied significantly by dpi, our results showed that the RNAi pathway (represented by gene *AGO*) was the major responder to JCV infection, as well as *Vago* (encodes an interferon-like factor), with expression levels increasing ~1000-fold at 10 dpi in both lines. We also found that the JAK/STAT pathway (represented by *hop*) demonstrated a 100-fold increase in expression at 14 dpi in the Wildtype (*Wolbachia*-free) line. AGO, Vago, and hop showed rising trends with increasing dpi (Fig 3). *MYD88* and *IMD* which represent Toll and IMD pathways, respectively, exhibited very little expression level change (< 10-fold) in response to JCV infection. When each gene was considered separately, we found all genes showed significant differences at different dpi (*MYD88*: $F_{3,247} = 41.71$, p < 0.001; *IMD*: $F_{3,247} = 23.20$, p < 0.001; *hop*: $F_{3,247} = 9.24$, p < 0.001; *AGO*:

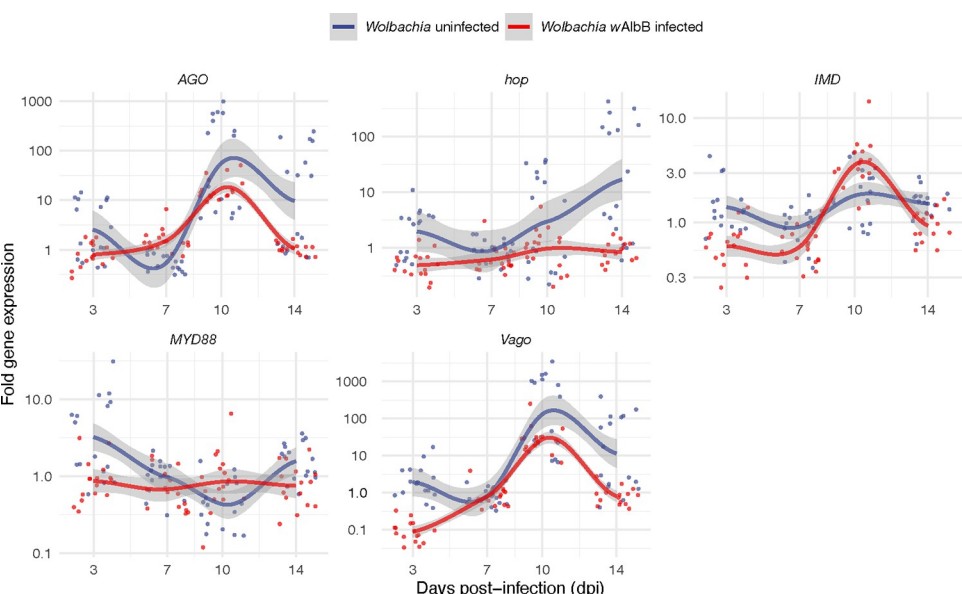

**Fig 3. Fold expression change for immunity genes in response to Jamestown Canyon Virus (JCV) infection via 2$^{-\Delta\Delta Ct}$ method.** Mosquitoes fed with mock-infected blood served as controls. Points are jittered. Smooth curves are plotted using the loess method to indicate the change trend, the gray area indicates 95% interval. *n* = 16 per treatment.

$F_{3,247}$ = 77.45, p < 0.001; *Vago*: $F_{3,247}$ = 78.57, p < 0.001). Specifically, *hop* ($F_{3,247}$ = 28.24, p = 0.018), *AGO* ($F_{3,247}$ = 4.97, p = 0.027), and *Vago* ($F_{3,247}$ = 16.02, p < 0.001) showed higher expression in the wildtype versus *Wolbachia*-infected lines. *Wolbachia* had no effect on the expression of gene *MYD88* ($F_{3,247}$ = 1.203, p = 0.274) and *IMD* ($F_{3,247}$ = 1.589, p = 0.209) after mosquito infection with JCV. In general, the impact of *Wolbachia* on mosquito immune gene expression was smaller in comparison to JCV (S2 Fig and S4 Table), reducing the expression of the JAK/STAT pathway and increasing the expression of the IMD pathway up to 5-fold dpi.

## Effect of Jamestown Canyon virus infection on *Wolbachia* density and female longevity

We quantified the relative density of *Wolbachia* for the whole body of *w*AlbB-infected mosquitoes at different days post-feeding with JCV-infected or mock-infected blood (S3 Fig). We found there was no effect of JCV infection ($F_{1,120}$ = 3.78, p = 0.054), a slight effect of dpi ($F_{3,120}$ = 2.78, p = 0.044), and no interaction ($F_{3,120}$ = 2.35, p = 0.08). *Wolbachia* significantly reduced survival (Fig 4) by pair-wise log-rank tests whether females had been fed with JCV ($\chi2$ = 14.5, df = 1, p < 0.001) or mock ($\chi2$ = 15.3, df = 1, p < 0.001). JCV infection had no effect on female survival for either *Wolbachia* infected ($\chi2$ = 0.4, df = 1, p = 0.52) and uninfected ($\chi2$ < 0.1, df = 1, p = 0.89) lines.

## Discussion

In this study, we tested the vector competence of *Ae. aegypti* for Jamestown Canyon virus (JCV) and investigated whether there is *Wolbachia*-mediated blocking against this negative-sense RNA virus. We demonstrated *Ae. aegypti*'s potential to be a highly competent vector given the following: 1) the infection prevalence and viral loads were very high, near 100% and $10^7$ viral copies/tissue, respectively, and 2), the virus disseminated rapidly, showing dissemination and presence in the saliva at only 3 days post-infectious feed. Additionally, the virus did

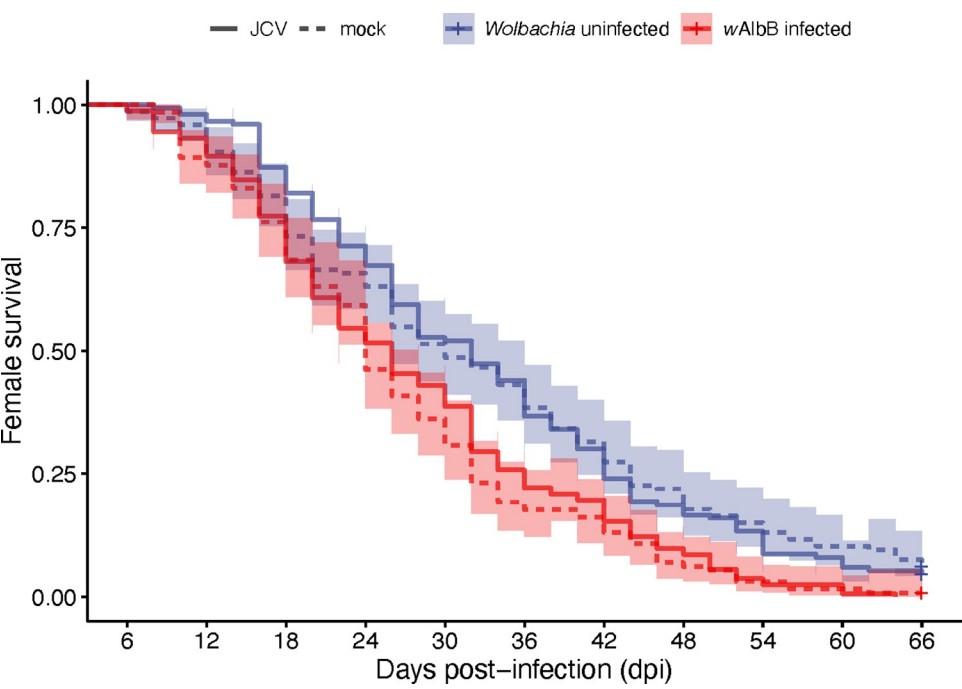

**Fig 4. Longevity of *Wolbachia*-infected and uninfected *Ae. aegypti* females post blood feed with either JCV or mock.** $n$ = 130–163 per treatment.

not reduce female mosquito longevity, leaving this aspect of vectorial capacity unaffected, and lifetime transmission potential high. In terms of the impact of *Wolbachia*, we found that *Wolbachia* only had a moderate ability to inhibit JCV, reducing virus prevalence and viral loads, largely very early during infection.

The emergence and spread of infectious diseases of zoonotic origin have become crucial issues in global human health in recent decades [44,45]. Our study shows that JCV, now circulating primarily in the upper Midwest [1,3] has the potential to become a much larger public health threat if its distribution collides with the expanding ranges of *Ae. aegypti* [8] and likely, its close relative, *Ae. albopictus*. Most of the current vectors of JCV are likely feeding on both humans and animals [2], directly facilitating human spillover infections. *Ae. aegypti*, in contrast, feeds with near-perfect fidelity on humans [46]. Its presence in the region would therefore not facilitate spillover but could drive a direct human-human transmission cycle if the virus were sufficiently common. *Ae. albopictus*, in contrast, is more flexible in its host range, depending on availability. It can feed exclusively on humans [46] or other mammals including deer [47] and hence could be a potential participant in both an enzootic and anthroponotic cycle. Interestingly, *Ae. albopictus* naturally harbors *Wolbachia* infection [35]. Our work suggests the symbiont may do little to limit the vector competence of wild *Ae. albopictus* for JCV if the virus establishes itself in an individual mosquito. JCV loads in the body of *Wolbachia*-infected *Ae. aegypti* were very high (averaging $10^7$ viral copies/disseminated tissues), following a starting total load of $10^6$ viral copies/midgut. These values are on the higher end of what is seen for similarly designed studies with DENV in *Ae. aegypti*, which is a natural vector (reviewed in [48]). Little is known about the range of JCV titers in the blood of infected humans or wild mammals. For the closely related LACV, titers resulting from artificial infections in deer, and other mammals ranged from $10^2$ to $10^5$/ml [49, 50]. Even if titers are low in human and wild hosts, *Ae. aegypti* is very effective at amplifying and disseminating JCV.

*Ae. aegypti* mounted a very strong immune response to JCV that peaked at 10 dpi, but with infection prevalence and viral loads continuing to remain high. This may suggest poor efficacy of the response, or more likely, a steady state between virus replication and anti-viral immune action. In the case of DENV infection where kinetics has been more closely examined, viral loads also plateau and remain high and constant [43] despite a well-described anti-DENV response [51]. The immune response largely consisted of the RNAi and JAK/STAT pathways. While negative-sense viruses do not make dsRNAs in sufficient quantities to trigger RNAi [52], regardless, this pathway has been shown in *Drosophila* [53] and in mosquitoes [54] to both respond to and limit negative-sense viruses. The parallel expression patterns of *AGO* and *Vago* in response to infection are likely due to molecular crosstalk. *Vago*, produced by the IMD pathway or *Wolbachia* infection, encodes an interferon-like factor that is also a mediator of both the JAK/STAT [33] and RNAi pathways [55]. Last, JCV infection did not appear to shorten mosquito lifespan, suggesting the virus is not cytotoxic in this novel host and that there are not extreme fitness tradeoffs for mounting the strong immune response seen here (reviewed in [56]). This is perhaps not surprising as LACV in its native vector *Aedes triseriatus* [57] has no discernable effect on fitness and the virus can live for very long periods in the mosquito, including in the ovaries. The latter is associated with transovarial transmission and over-wintering of the virus in northern climates.

*Wolbachia*-mediated blocking is known to be a multifaceted blocking trait [16], whose composition may vary depending on the *Wolbachia* strain, virus species/genotype, and vector species/genotype. One interesting feature of our findings is that virus prevalence and load appear to be disconnected, as if two independent traits. In contrast, for dengue virus, low prevalence and low viral loads together are characteristic of *Wolbachia*-mediated blocking [58][58]. With JCV, *Wolbachia*-infected mosquitoes appear better at preventing infection, rather than controlling it, once initiated. This may suggest early events are of most importance for *Wolbachia*-mediated blocking of negative strand viruses. Previous studies have suggested that viral RNA early in infection is the primary target for blocking [59], possibly recognized by the immune system [17, 60], but that *Wolbachia*-induced changes in mosquito physiology involving lipid trafficking [61][61], autophagy [61,62], and reactive oxygen species [17], etc can also play a role. Dissecting the relative contributions of these multiple components has been challenging in a tripartite organismal system, with no ability to genetically modify *Wolbachia* and difficulties in modifying DENV. The presence of *Wolbachia*-mediated blocking of JCV in both tissues and saliva early in infection offers new comparative potential for dissecting blocking mechanisms relative to positive-sense viruses. The innate immune pathways including RNAi [19], IMD (including *Vago*), and Toll are involved with *Wolbachia*-mediated blocking of positive-sense viruses [18]. Relative to JCV infection, however, *Wolbachia* only moderately induced immune gene expression. Furthermore, during co-infection, *Wolbachia* appeared also to dampen the immune induction by JCV at some time points (see *AGO*, *vago*, and *hop*, as well as *MYD88* and *IMD* at 3 dpi) suggesting immune activation is unlikely to be at play here. The strength of these interpretations is, however, somewhat limited by the sample sizes for the gene expression.

We propose a hypothetical model for why *Wolbachia*-mediated blocking may be less able to control JCV than positive-sense viruses. LACV is known to cause a dramatic shut-off of host protein synthesis in BHK cells, and then subsequently, viral protein synthesis in the later stages of infection [63]. This strategy is thought to protect viruses from attack by host antiviral factors. Mosquito cells infected with LACV continue to divide and behave normally despite infection and so appear unaffected. *Wolbachia* infection has a related effect on host protein synthesis, reducing it by 23% in *Drosophila* JW18 cells [64], but not shutting it off as per LACV. Positive-sense viruses, without an mRNA intermediate, would be dependent on host polymerase and their replication activity would slow. Negative-sense viruses, utilize their own

polymerase to produce mRNA copies of their genomes. Because the activity level of this polymerase is highest when low amounts of viral protein are present, *Wolbachia* induced reductions in protein synthesis should instead lead to the overproduction of JCV mRNAs. Second, other members of the *Peribunyaviridae*, in mammalian cells, have been shown to remove the 5' terminus of their genome post-transcriptionally to prevent detection by receptors that trigger interferon activity [65]. There may be parallel stealth activities in the vector to avoid triggering the action of the interferon-like factor Vago [66]. This is difficult to assess, however, with many aspects of the insect antiviral pathways still undefined including key receptors, and modes of action. When examining *Wolbachia* infection alone (S2 Fig), the strongest immune activation was for *Vago* early during infection. The impact of *Wolbachia's* activation of *Vago* may be greater for DENV and other positive sense viruses where viral infection induction of the gene is much less strong than for JCV [32]. There are alternate explanations, including because of low JCV levels early in infection, there are insufficient PAMPs to trigger antiviral pathways, or that JAK/STAT may be necessary, but insufficient to control viruses as has been shown previously [67]. We finish by saying that studying the *Wolbachia*-mediated blocking seen here early during infection may reveal the functional importance of more universal aspects of blocking not specific to virus type.

In summary, we show that *Ae. aegypti* has the potential to be a very effective vector of JCV in the future, with the mosquito's range shifting northward [8], and JCV being present beyond the Midwest. Future studies should explore the vector competence of *Ae. albopictus* for JCV, which given its more temperate range, may be a more imminent threat. Additionally, characterizing JCV titers in wild mammals and humans may help to provide context for the vector competence seen here. We also show some evidence of *Wolbachia*-mediated blocking of a negative-sense virus inside the mosquito, whereas previous *in vitro* studies with other viruses found no evidence of blocking [23,24]. Evidence of weak blocking suggests that JCV may serve as a good comparator for teasing out both aspects of blocking that may vary by virus sense-specific biology and indeed those that are more universal. We propose two hypothetical scenarios that may be explored experimentally *in vitro* and *in vivo*. This work should revive interest in studying negative-sense viruses in mosquitoes as a model for *Wolbachia*-mediated blocking. One specific comparative study that may be of value would be to assess the expression of downstream genes induced by Vago in response to negative versus positive sense viruses.

## Supporting information

**S1 Table. Jamestown Canyon virus loads in subsample of whole-body mosquitoes immediately after infectious feeding (day 0) to confirm that mosquitoes received a substantial virus dose (~10$^6$).** *W-* = wildtype line, *W+* = *Wolbachia* infected. Use of mosquitoes for different experiments is listed.
(DOCX)

**S2 Table. Primers used for gene expression.** *rps17* is the reference gene.
(DOCX)

**S3 Table. Pair-wise comparisons for viral load between *Wolbachia w*AlbB-infected and uninfected *Ae. aegypti* females.**
(DOCX)

**S4 Table. Pairwise comparisons for gene expression between *Wolbachia w*AlbB-infected and uninfected *Ae. aegypti* females.**
(DOCX)

**S1 Text. qRT-PCR assays for Jamestown Canyon virus quantification.**
(DOCX)

**S1 Fig. Vector competence for the second replicate.** (A) Jamestown Canyon virus (JCV) infection prevalence in adult female mosquito saliva and legs at 3, 7, 10, and 14 days post-infection (dpi). *n* = 30 per treatment. In both abdomen and head & thorax, there were significant effects for both *Wolbachia* infection (abdomen: $z = -2.63$, $p = 0.008$; head & thorax: $z = -3.14$, $p = 0.002$) and dpi (abdomen: $z = 4.50$, $p < 0.001$; head & thorax: $z = 3.66$, $p < 0.001$) on infection prevalence. (B) For viral load, in either abdomen and head & thorax, there was no significant impact of *Wolbachia* infection (abdomen: $t = 0.22$, $p = 0.83$; head & thorax: $t = 1.94$, $p = 0.053$) or dpi (abdomen: $t = 1.85$, $p = 0.066$; head & thorax: $t = -0.97$, $p = 0.33$) but a significant interaction was found for head & thorax (abdomen: $t = 0.22$, $p = 0.83$; head & thorax: $t = 2.67$, $p = 0.008$).
(DOCX)

**S2 Fig. Fold-change expression for immune genes between *Wolbachia* *w*AlbB infected and uninfected mosquitoes after a mock-infected blood meal (no virus).** The maximum change of gene expression levels in response to *Wolbachia* infection was less than 10-fold for all genes at all dpi. *n* = 16 per treatment. Significant differences were found between *Wolbachia* infected and uninfected lines ($F_{1,600} = 10.39$, $p = 0.001$), among different genes ($F_{4,600} = 27.00$, $p < 0.001$), and among different dpi ($F_{3,600} = 13.41$, $p < 0.001$). When genes were considered separately, gene *AGO* ($F_{1,120} = 8.67$, $p = 0.004$), *hop* ($F_{1,120} = 9.78$, $p = 0.002$), and *Vago* ($F_{1,120} = 59.47$, $p < 0.001$) were affected by *Wolbachia* infection, but not for gene *MYD88* ($F_{1,120} = 3.81$, $p = 0.053$) and *IMD* ($F_{1,120} = 0.20$, $p = 0.66$).
(DOCX)

**S3 Fig. Relative *Wolbachia* density in whole *w*AlbB-infected *Ae. aegypti* at various days post blood feed with either JCV or mock.** *n* = 16 per treatment.
(DOCX)

## Acknowledgments

The authors would like to thank Heather Engler for assistance with mosquito rearing.

## Author Contributions

**Conceptualization:** Meng-Jia Lau, Heverton L. C. Dutra, Elizabeth A. McGraw.

**Formal analysis:** Meng-Jia Lau, Matthew J. Jones.

**Funding acquisition:** Elizabeth A. McGraw.

**Investigation:** Meng-Jia Lau, Heverton L. C. Dutra, Brianna P. McNulty, Anastacia M. Diaz, Fhallon Ware-Gilmore.

**Supervision:** Elizabeth A. McGraw.

**Writing – original draft:** Meng-Jia Lau, Elizabeth A. McGraw.

**Writing – review & editing:** Meng-Jia Lau, Heverton L. C. Dutra, Matthew J. Jones, Brianna P. McNulty, Anastacia M. Diaz, Fhallon Ware-Gilmore, Elizabeth A. McGraw.

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
