## [Decision Letter · Decision Letter 0]

27 Jul 2023

Dear Prof McGraw,

Thank you very much for submitting your manuscript "Jamestown Canyon Virus is transmissible by Aedes aegypti and is only moderately blocked by Wolbachia co-infection" for consideration at PLOS Neglected Tropical Diseases. As with all papers reviewed by the journal, your manuscript was reviewed by members of the editorial board and by several independent reviewers. The reviewers appreciated the attention to an important topic. Based on the reviews, we are likely to accept this manuscript for publication, providing that you modify the manuscript according to the review recommendations. 

Sincerely,

Doug E Brackney, PhD

Academic Editor

Abdallah Samy

Section Editor

Reviewer's Responses to Questions

**Key Review Criteria Required for Acceptance?**

**Methods**

-Are the objectives of the study clearly articulated with a clear testable hypothesis stated?

-Is the study design appropriate to address the stated objectives?

-Is the population clearly described and appropriate for the hypothesis being tested?

-Is the sample size sufficient to ensure adequate power to address the hypothesis being tested?

-Were correct statistical analysis used to support conclusions?

-Are there concerns about ethical or regulatory requirements being met?

Reviewer #1: The manuscript is well-written, and the experiments are well-controlled. The following statements are well supported by the data presented. However some changes (suggested below) can significantly improve the interpretation and impact of the data. Most experiments described include adequate controls. Most experiments have been designed to address the stated objectives. Suggestions have been made below in the summary section.

Reviewer #2: (No Response)

Reviewer #3: All methods seem valid and standard. Methods address the questions asked. Sample sizes are a bit low in some instances. Parts of Fig. 2, Fig. 3, Fig. 4, Fig. S2 had sample sizes of 16-17 mosquitoes. Ideally these numbers would be increased to ~25-30. 

Wolbachia infected and uninfected mosquitoes have different genetic backgrounds, correct? This is likely ok but make this more clear in the methods section and perhaps the discussion.

See line by line comments for minor edits.

**Results**

-Does the analysis presented match the analysis plan?

-Are the results clearly and completely presented?

-Are the figures (Tables, Images) of sufficient quality for clarity?

Reviewer #1: See response in the summary below.

Reviewer #2: (No Response)

Reviewer #3: Results are mostly clear with some minor edits below. 

Line 139-140: Seems out of place? I think this is in response to the bimodal distribution of data points in 1B but it seems out of order here and does not get revisited later. 

Fig. 1: It should be clearer from the figure legend/caption that this is rep 1 data only. 

Line 184-185: These p values seem like they may not be significant? “Viral load in legs (Fig. 2B) was

significant for both dpi (t = - 3.64, p < 0.49) and Wolbachia infection (t = 6.07, p < 0.49).” It’s possible these are <0.05 but it seems that this could be more specific if so?

Line 188: These results showing increased viral load in the legs but a decreased JCV prevalence with Wolbachia should be brought up in the discussion. Why is this happening? Has this been shown before? Does this speak to the mechanism of Wolbachia-mediated blocking? 

Fig. 5: The impacts of this result should be explained more, perhaps in the discussion. Is it normal for introduced Wolbachia infections to lead to decreased lifespan? Does this impact the viral transmission window?

Fig. S3: This needs more explanation and a caption. What is AFM? What is wBM?

**Conclusions**

-Are the conclusions supported by the data presented?

-Are the limitations of analysis clearly described?

-Do the authors discuss how these data can be helpful to advance our understanding of the topic under study?

-Is public health relevance addressed?

Reviewer #1: Included in the summary.

Reviewer #2: (No Response)

Reviewer #3: Conclusions are appropriate and supported by the data presented. Minor comments below.

Line 253-255: This comparison needs to be clarified. Ae. albopictus is still a capable vector of many viruses despite its natural Wolbachia infection. There’s no reason to believe these naturally occurring infections would block JCV however, if an additional non-native Wolbachia strain is added to Ae. albopictus, it may behave in a similar way to the Ae. aegypti in this paper and offer limited blocking. 

Reading this again it’s possible I’m not understanding and this is actually citing other work that looked at JCV in Ae. albopictus-in which case add the citation and clarify!

Address the low sample sizes in several figures. This limits some of the conclusions that can be made.

**Editorial and Data Presentation Modifications?**

Reviewer #1: The authors should define the terms “viral prevalence” and “viral load” in the results. These two measurements should also be clearly outlined and distinguished in the materials and methods section.

Reviewer #2: (No Response)

Reviewer #3: The paper has some minor grammar and editing errors that can be easily fixed with another pass through.

**Summary and General Comments**

Reviewer #1: This study by Lau et al. is timely given the increasing prevalence of JCV in North America and deployment of Wolbachia-colonized Aedes aegypti mosquitoes to prevent arbovirus transmission. The endosymbiont Wolbachia has been shown to prevent transmission of plus-sense RNA viruses via a plethora of cell-autonomous mechanisms. However, evidence of Wolbachia’s ability to limit infection by negative-sense RNA viruses in vivo is limited and needs to be explored to a greater degree. The authors are therefore justified in their rationale for testing the pathogen blocking ability of Wolbachia against negative-sense RNA viruses by using JCV. The manuscript is well-written, and the experiments are well-controlled. The following statements are well supported by the data presented. However some changes (suggested below) can significantly improve the interpretation and impact of the data. 

1) Presence of Wolbachia has a small, but significant effect on JCV prevalence and viral load early during infection. Figure 1, 2, S1, Table S1: Collectively, these data show that Aedes aegypti can function as a competent JCV vector. The authors have done an appreciable job at reporting the results of two experiments independently. The differences observed between these two experiments highlight the variability in the pathogen blocking phenotype. Most pronounced differences in viral prevalence/load between mosquitoes with and without wAlb are at 3 and 14dpi. Can the authors comment on whether this relates to a delay in establishing infection and faster viral clearance?

2) The authors quantify the expression of a limited set of immune genes encompassing major arthropod antiviral pathways and find that the presence of the endosymbiont alone elevates the expression of vago, a secreted protein that has been shown to function like an interferon-like molecule. The data presented in S2 shows that wAlb is associated with elevated vago expression. This seems to be an important finding and the authors should consider including this as a primary figure. Vago has been suggested to function as an interferon-like molecule. Therefore, it would be worthwhile to perform comparative transcriptomics of wAlb colonized Aedes aegypti infected with JCV and a plus-strand RNA virus like DENV/YFV or CHIKV. Based on previous studies, one would hypothesize that Vago induction would lead to the downstream induction of the JAK/STAT pathway (1, 2). In the present study, the authors investigate the effect of JCV and Wolbachia on the expression of a handful of candidate host immune genes, including the JAK/STAT pathway gene hopscotch. 

However, it would be better to look at the expression of downstream genes that are responsive to the activation of the JAK/STAT like vir-1 (3) to get a functional readout of vago-induced JAK/STAT activation. 

3) Expression of most antiviral pathways seems to be correlated with virus prevalence at early and later time points post infection. Figure 3, S2: Collectively, figure 3 shows that the level of immune gene induction is correlated with JCV prevalence in mosquitoes with and without wAlb, in that differences are pronounced at 3 and 14dpi. 

In this regard, including a figure showing the correlation between immune induction and viral RNA prevalence would be worthwhile. This will also bolster the statement made in the discussion (lines 296 – 298). The alternate explanation of which is that the insect immune response is not dampened by Wolbachia but rather not induced given lower infection levels.

4) Presence of Wolbachia, but not JCV impose a fitness cost to the vector. 

5) JCV infection does not have any observable impact on endosymbiont titer. While important, this data can be moved to the supplemental section.

Taken together, these data present a comprehensive assessment of how Wolbachia influences in vivo JCV prevalence. However, given the data presented, and the lack of previous data in related infection models, I believe there is not enough information in this present study that the authors can use to propose hypothetical models of JCV blocking, which require hypothetical in vivo transcriptomic/proteomic data. Consequently, a significant portion of the discussion (lines 300 – 322) need to be revisited. In particular, the hypothesis that Wolbachia-induced translational blockade may lead to over production of JCV mRNA is unsubstantiated given that the authors do not identify changes in JCV mRNA levels in the presence of Wolbachia. The authors also hypothesize that negative-sense viruses evade detection by RLRs by modifying the 5’ end of their genomes, which prevent the induction of IFN/vago expression, explaining the absence of immune induction by JCV. It is also likely, that given reduced JCV levels at 3dpi, that there are not enough PAMPs to trigger antiviral pathways. But does JCV evade the action of insect immune pathways that are triggered by pre-existing vago in Wolbachia-colonized mosquitoes (Figure S2)? It would seem not, given that viral prevalence/loads are reduced early in infection (Figure 1,2, S1A). An alternate model would be this: Previous work suggests that the JAK/STAT pathway is required but is not sufficient for antiviral response in insects (3). It is therefore possible, that the antiviral response to JCV in Wolbachia-colonized mosquitoes is incomplete and viral loads recover later in infection. 

1. Paradkar, Prasad N., Trinidad, Lee, Voysey, Rhonda, Duchemin, Bernard, and Peter J. Walker. "From the Cover: Secreted Vago restricts West Nile virus infection in Culex mosquito cells by activating the Jak-STAT pathway." Proceedings of the National Academy of Sciences of the United States of America 109, no. 46 (2012): 18915-18920. Accessed July 18, 2023. https://doi.org/10.1073/pnas.1205231109.

2. Gao J, Zhao BR, Zhang H, You YL, Li F, Wang XW. Interferon functional analog activates antiviral Jak/Stat signaling through integrin in an arthropod. Cell Rep. 2021 Sep 28;36(13):109761. doi: 10.1016/j.celrep.2021.109761. PMID: 34592151.

3. Dostert C, Jouanguy E, Irving P, Troxler L, Galiana-Arnoux D, Hetru C, Hoffmann JA, Imler JL. The Jak-STAT signaling pathway is required but not sufficient for the antiviral response of drosophila. Nat Immunol. 2005 Sep;6(9):946-53. doi: 10.1038/ni1237. Epub 2005 Aug 7. PMID: 16086017.

Reviewer #2: Lau et al. investigated the vector competency of Aedes aegypti to transmit Jamestown Canyon virus and the effect of Wolbachia on the replication of JCV. The authors showed that Ae. aegypti is a highly competent vector for JCV. While positive sense-RNA viruses are strongly blocked by Wolbachia, the study shows that blocking of JCV occurs only early in infection. JCV infection did not affect female survival, while Wolbachia reduced their longevity. Although the effect of Wolbachia on negative-sense RNA viruses has been shown in cell lines (no effect), this hasn’t been tested in mosquitoes. Therefore, the present study advances our knowledge of Wolbachia effects on negative-sense RNA viruses. The manuscript is generally well-written, and the results support the conclusions.

Minor comments:

Line 22, 47, 133, 135, 190, 231, 353, 675, Fig. 3 line 1: it is more correct to write the virus name as Jamestown Canyon virus. Also, to be consistent with the rest of the manuscript.

Line 64: Orthobunyavirus genus belongs to the family Peribunyaviridae of the order Bunyavirales. Bunyaviridae was disestablished 2-3 years ago.

Line 82: Ae. aegypti

Line 92: this statement is not totally correct because we know that Wolbachia doesn’t block DNA or negative-sense RNA viruses. I would moderate the sentence.

Line 106: references?

Line 124: negative-sense RNA viruses

Line 170: two instances of “average”

Lines 183 and 187: I am finding results between these two lines confusing. While it is earlier mentioned that viral load in saliva was not affected by Wolbachia infection, later it is mentioned 50% reduced viral load due to Wolbachia infection. Please clarify.

Line 245: its close relative

Line 271: there is at least one example from mosquitoes showing RNAi response to a negative-sense RNA virus: mSphere. 2017 May 3;2(3):e00090-17

Line 276: aren’t should be are not

Lines 301-310: I am not sure if the hypothesis described is totally relevant. If protein synthesis in the presence of Wolbachia is inhibited in general, it should also affect positive-sense RNA viruses because they produce a large number of positive-sense viral RNAs which are used similar to mRNA for protein synthesis, which include the RdRp protein.

Lines 339 and 342: W- and W+ aren’t used anywhere else in the manuscript. You may delete these abbreviations.

Line 339-342: are these lines genetically identical? They don’t seem to be. Usually, a tet-treated line is used as the Wolbachia-free line. 

Line 354 and 355: media should be medium

Lines 395-397: rps17 primer sequences are already listed in Table S4 and Wolbachia primers can also be added to the table and sequences removed from the text. Also, it is more useful to mention the name of the Wolbachia gene used rather than just Wolbachia for the primers.

Line 400: S1 Text?

Line 405: qPCR

Line 424: shouldn’t Table S4 be S2? Table S4 lists primers.

Line 640: no need to repeat Jamestown Canyon virus (JCV)

Line 682: DNase, RNase

Line 688: Taq

Table S3: please define AFM and wBM in the legend or under the table.

Reviewer #3: While the sample size of is a bit low for some parts of the study (selected parts of Fig. 2, Fig. 3, Fig. 4, Fig. S2 had sample sizes of 16-17 mosquitoes) this study offers new information about the role and mechanisms of Wolbachia-mediated pathogen blocking as well as the potential for Ae. aegypti to serve as a vector of Jamestown Canyon virus. The conclusions are appropriate and within the scope of the data gathered. While a larger sample size would greatly strengthen this study, conclusions are moderate and point to future experiments that could build upon this data.

PLOS authors have the option to publish the peer review history of their article (what does this mean?). If published, this will include your full peer review and any attached files.

Reviewer #1: No

Reviewer #2: No

Reviewer #3: No

Figure Files:

Data Requirements:

Reproducibility:

References

---

## [Editor Report · Decision Letter 1]

22 Aug 2023

Dear Prof McGraw,

We are pleased to inform you that your manuscript 'Jamestown Canyon Virus is transmissible by Aedes aegypti and is only moderately blocked by Wolbachia co-infection' has been provisionally accepted for publication in PLOS Neglected Tropical Diseases.

Best regards,

Doug E Brackney, PhD

Academic Editor

Abdallah Samy

Section Editor

---

## [Editor Report · Acceptance letter]

30 Aug 2023

Dear Prof McGraw,

We are delighted to inform you that your manuscript, "Jamestown Canyon Virus is transmissible by *Aedes aegypti* and is only moderately blocked by Wolbachia co-infection," has been formally accepted for publication in PLOS Neglected Tropical Diseases.

Best regards,

Shaden Kamhawi

co-Editor-in-Chief

Paul Brindley

co-Editor-in-Chief
